# Phytochemical Screening, Anti-Inflammatory, and Antidiabetic Activities of Different Extracts from *Caralluma edulis* Plant

**DOI:** 10.3390/molecules27165346

**Published:** 2022-08-22

**Authors:** Maria Khan, Zahid Manzoor, Muhammad Rafiq, Shaukat Hussain Munawar, Muhammad Yasir Waqas, Hamid Majeed, Syed Zahid Ali Shah, Riaz Hussain, Hafiz Iftikhar Hussain, Tehreem Tahir, Katarzyna Kotwica-Mojzych, Mariusz Mojzych

**Affiliations:** 1Department of Physiology and Biochemistry, Faculty of Bio-Sciences, Cholistan University of Veterinary and Animal Sciences, Bahawalpur 63100, Pakistan; 2Department of Pharmacology and Toxicology, Faculty of Bio-Sciences, Cholistan University of Veterinary and Animal Sciences, Bahawalpur 63100, Pakistan; 3Department of Food Science and Technology, Faculty of Bio-Sciences, Cholistan University of Veterinary and Animal Sciences, Bahawalpur 63100, Pakistan; 4Department of Pathology, Faculty of Veterinary and Animal Sciences, The Islamia University of Bahawalpur, Bahawalpur 63100, Pakistan; 5Department of Pathology, Faculty of Veterinary Sciences, Cholistan University of Veterinary and Animal Sciences, Bahawalpur 63100, Pakistan; 6Institute of Biochemistry, Biotechnology and Bioinformatics, Faculty of Science, The Islamia University of Bahawalpur, Bahawalpur 63100, Pakistan; 7Laboratory of Experimental Cytology, Medical Faculty, Medical University of Lublin, Radziwiłłowska 11, 20-080 Lublin, Poland; 8Department of Chemistry, Siedlce University of Natural Sciences and Humanities, 08-110 Siedlce, Poland

**Keywords:** *Caralluma edulis*, phytochemical screening, anti-inflammatory, antidiabetic, rabbits, antioxidant, GC-MS analysis

## Abstract

The plant *Caralluma edulis* is traditionally used against diabetes and inflammatory conditions in Pakistan. This study was designed to provide scientific validation of the traditional use of *Caralluma edulis*. Phytochemicals were extracted from the plant by different solvents (distilled water, methanol, ethanol, and acetone) using the Soxhlet’s extraction method. Bioactive compounds were detected by gas chromatography–mass spectrometry (GC-MS). The in vitro anti-inflammatory activities (albumin denaturation, membrane stabilization, and proteinase inhibition) and antioxidant capacity (DPPH scavenging activity, FRAP reducing activity) of different extracts from *Caralluma edulis* were assessed. The antidiabetic potential of *Caralluma edulis* plant extracts was determined in acute and subacute diabetic rabbit models. Oxidative stress and enzymatic antioxidant status were also estimated in MDA, CAT, and SOD levels. Results showed that the methanol extract yielded the highest contents of phenolics, flavonoids, alkaloids, and terpenoids. The in vitro anti-inflammatory activity and antioxidant potential of the methanol extract were the highest among the tested solvents. The tested extracts did not show any remarkable antidiabetic activity in the acute diabetic model. However, all tested extracts demonstrated antidiabetic potential in the subacute diabetic model. No adverse effect was observed at the tested dose (200 mg/kg) of *Caralluma edulis* extracts in experimental animals. It is concluded that methanol is the key solvent for extracting bioactive compounds from *Caralluma edulis*. The plant can be used against inflammatory disorders and may prove a potential candidate for drug development. Long-term use of *Caralluma edulis* at the tested dose (200 mg/kg) showed antidiabetic properties in the animal model.

## 1. Introduction

*Caralluma edulis* (Apocynaceae) is a succulent herb found in many parts of the South Asian region [1]. It has been used in folk medicines to manage specific diseases such as diabetes mellitus (DM), cancer, tuberculosis, snake and scorpion bites, scabies, skin rashes, fever, and inflammation [2]. *Caralluma edulis* is an edible plant, and many people use it as a part of their food. Recent investigations demonstrated high antioxidants in the methanolic plant extract and many specific bioactive compounds such as glycosides, flavonoids, alkaloids, quinones, saponins, and phenols [3].

Inflammation is a body’s immune response to harmful stimuli such as toxic compounds, pathogens, damaged cells, injuries, or irritation. It may be associated with certain diseases, including arthritis, asthma, cancer, diabetes, autoimmune diseases, and neurodegenerative diseases [4]. Treatment of inflammatory disorders by conventional drugs is usually associated with severe side effects; therefore, alternative anti-inflammatory drugs from natural sources have recently gained much attention [5].

DM is one of the prevailing endocrine disorders. In 2021, it was estimated that 537 million people had diabetes, around 10.5% of the population. In addition, health expenditures due to diabetes were USD 966 billion in 2021 [6]. DM affects the carbohydrate, lipoprotein, and lipid metabolisms that may result in hyperglycemia and many other disorders, such as atherosclerosis, hypertension, hyperlipidemia, and hyperinsulinemia [7]. It is well documented that hyperglycemia induces oxidative stress, which results in the production of free radicals that play a major role in the development and pathogenesis of DM [8]. In addition, an increased amount of free radical generation damages the cell membrane through the processes of lipid peroxidation and protein glycation and weakens the antioxidant defence mechanism of the cell [9].

Malondialdehyde (MDA) is a degradative product of the peroxidation of polyunsaturated fatty acids and considered the indicator of oxidative stress [10]. Reactive oxygen species (ROS) induce chemical modification in all cellular components and produce lipid peroxidation. Hence, oxidative stress also causes DM. As a result, lipid peroxidation is an important cause of DM [11]. The scavenger enzymes such as catalase, glutathione peroxidase, and superoxide dismutase provide protection against ROS. Catalase neutralizes H_2_O_2_ by catalytically converting it to water and oxygen. When catalases are deficient, pancreatic islet cells are more susceptible to the excessive formation of ROS and oxidative stress, which leads to pancreatic islet dysfunction and overt DM [12]. The plants containing antioxidant compounds can protect β-cells from ROS and, therefore, can prevent DM induced by ROS [13]. Therefore, it is pertinent to explore the traditional medicinal plants that can be utilized for lowering blood glucose levels.

The extraction process is the primary technique used to obtain these bioactive compounds from the biomass material. This analytical tool aims to achieve the highest amount of the target compounds from the material [14]. The extraction technique and solvent affect the bioactive compounds’ extraction yield and the resulting extract’s pharmacological action [15]. There is a large diversity in the plants’ bioactive compounds, and their extraction yield is affected by different solvents. The selection of the solvent for the extraction process depends on the isolation of the desired compounds [16]. Therefore, it seems very difficult to recommend a suitable solvent for individual plant materials. Previous studies on *Caralluma edulis* demonstrated the screening of bioactive compounds in a single solvent. However, the effect of different solvents on the extraction yield of bioactive compounds from *Caralluma edulis* and their pharmacological activities have not been previously reported. The present study investigated the impact of different solvents (distilled water, methanol, ethanol, and acetone) on the extraction yield and content of flavonoids, alkaloids, terpenoids, and phenolics. The antidiabetic, antioxidant, and anti-inflammatory activities of *Caralluma edulis* were also examined.

## 2. Materials and Methods

### 2.1. Chemicals

All the chemicals used in the experiment were research grade and purchased from Sigma-Aldrich, Ascent, Singapore.

### 2.2. Collection and Identification of Plant Material

*Caralluma edulis* was collected from the Cholistan desert district Bahawalpur (Punjab Province, Pakistan) and was identified by the Botany Research Laboratory, Islamia University, Bahawalpur, Pakistan. The plant material was washed under running tap water and dried at room temperature for 15 days. After complete dryness, the plant material was ground in a mill (Eapmic, Shenzhen, China). The obtained powdered material was used for further experiments.

### 2.3. Preparation of Plant Extracts

Different solvents (distilled water, methanol, ethanol, and acetone) were used to investigate their impact on extraction ratio and the content of alkaloids, flavonoids, phenolics, and terpenoids. *Caralluma edulis* extract was prepared by the method described by Do et al. [17] with slight changes. One gram of plant material was immersed in different solvents (distilled water, methanol, ethanol, and acetone) by a ratio of 1:20 (*w*/*v*). These mixtures were kept at 60 °C for 24 h and then homogenized using a homogenizer (IKA, Breisgau-Hochschwarzwald, Germany) for 4 h. The obtained mixtures were then filtered and concentrated at 60 °C by a rotary evaporator (RE-5000A, Henan Lanphan Technology Co., Ltd., Zhengzhu, China). The samples were freeze-dried for 24 h and stored at 4 °C before further use. The experiments were repeated in triplicate.

### 2.4. GC-MS Analysis

The methanolic extract of *Caralluma edulis* was filtered with polymeric solid phase extraction (SPE) column and analyzed in GC-MS (QP2020, Shimadzu, Kyoto, Japan). Helium (99.99%) was used as carrier gas at a 1 mL/min constant flow. Sample injection (0.5 μL) was employed (split ratio of 10:1). GC column (elite-1) fused with silica capillary column (30 mm × 0.25 mm ID × 1 μM df, composed of 100% dimethyl polysiloxane) was used. The heated filament produced electrons (70 eV). The injector temperature was 250 °C, and the ion source temperature was 280 °C. The oven temperature was programmed from 110 °C, increasing 10 °C/min to 290 °C. Mass spectra were taken at 70 eV. The scanning interval was 0.5 s, and fragments were made from 45 to 450 Da. Total GC running time was 18 min.

#### Identification of Components

The database of the National Institute Standard and Technology (NIST) have more than 62,000 patterns and was used for comparing the spectrum obtained by GC-MS. Unknown components were identified by comparing the known components’ names, structure, and molecular weight stored in the NIST library.

### 2.5. Determination of Extraction Yield

The following equation estimated the extraction yield (%)
Extraction yield (%) = weight of the extract after evaporating solvent and freeze drying/dry weight of the sample × 100(1)

### 2.6. Determination of Total Phenolic, Alkaloid, Terpenoid, and Flavonoid Contents

Absolute ethanol was used as a diluting solvent to determine the flavonoid, phenolic, terpenoid, and alkaloid contents in different solvents extracted. One gram of freeze-dried extract of each solvent was mixed with absolute ethanol by a ratio of 1:10 (*w*/*v*) for the determination of these bioactive compounds.

#### 2.6.1. Total Phenolics (TPC)

The total phenolic contents were determined by the modified Folin–Ciocalteu assay [18]. Briefly, 1 mL of each extract was mixed with 5 mL of 10% Folin–Ciocalteu reagent. Then 4 mL of 2% Na_2_CO_3_ was added to the mixture. The control sample contained a reagent with absolute ethanol only and no plant extract. All the samples were incubated at room temperature for 2 h. The absorbance of all samples was then measured at 765 nm using the UV–visible spectrophotometer (Jasco, Portland, OR, USA). The phenolic contents of the samples were determined by establishing the calibration curve for gallic acid (0–100 μg/mL) and were expressed as the mg gallic acid equivalent (GAE) per gram of extract (dry weight). The calibration curve equation was y = 0.0073x − 0.0128, where R^2^ = 0.9992.

#### 2.6.2. Total Flavonoids (TFC)

Total flavonoid contents of *Caralluma edulis* were calculated by the modified aluminum chloride colorimetric method [19]. The mixture containing 0.5 mL of AlCl_3_ (5%), 0.5 mL of potassium acetate solution (1 M), and 2 mL of each sample extract was incubated at 25 °C for 15 min. The control sample contained absolute ethanol. UV–visible spectrophotometer (Jasco, Portland, OR, USA) was set at 415 nm to measure the absorbance of all samples. The reference standard for flavonoid estimation was quercetin. The result was determined as mg of quercetin equivalent (QE) per gram of extract (dry weight). The calibration curve equation was y = 0.0265x + 0.154, where R^2^ = 0.996.

#### 2.6.3. Total Alkaloids (TAC)

Total alkaloids were estimated by the colorimetric method [20]. In total, 1 mL of sample extract was taken, and its pH was adjusted to neutral by the consecutive washing of 10 mL of chloroform. The plant extract was then mixed with 5 mL of bromocresol green solution. A total of 5 mL of phosphate buffer was added. The resultant solution was vigorously shaken with chloroform in a flask. UV–visible spectrophotometer (Jasco, Portland, OR, USA) was set at 470 nm to measure the absorbance of all samples. Atropine was used as a reference. The result was expressed as mg of atropine equivalent (AE) per gram of extract (dry weight). The calibration curve equation was y = 0.0032x + 0.027, where R^2^ = 0.994.

#### 2.6.4. Total Terpenoids (TTeC)

Total terpenoids in *Caralluma edulis* were calculated by the method in [21]. The crude plant material was mixed in absolute ethanol and allowed to stand for 24 h. The mixture was then filtered. Petroleum ether was then added to the obtained filtrate. Total terpenoids were measured in the ether extract by the following equation.
Total terpenoids (%) = Final weight of the sample − Initial weight of the extract/Weight of the sample × 100(2)

#### 2.6.5. Quantification of Phytochemicals

The concentration of some important phytochemicals (campesterol, β-sitosterol, (*E*)-stilbene, gallic acid, pentadecanoic acid) were measured by UV–visible spectrophotometer (Jasco, Portland, OR, USA) at different wavelengths. These compounds have demonstrated antioxidant, anti-inflammatory, and antidiabetic properties in recent studies [22,23,24]. 

### 2.7. Acute Toxicity Studies

For the acute toxicity study, 30 rabbits weighing between 1.5–2 kg were selected. The animals were divided into six groups, each containing five rabbits. Group-1 served the control group, whereas group-2 to group-6 were offered *Caralluma edulis* plant extract suspended in water at a dose of 50, 100, 150, 200, and 250 mg/kg to each animal for 14 days. Oral acute toxicity studies were performed according to the guidelines laid down by OECD/OCDE 408. The animals were closely observed for any behavioral changes, adverse symptoms, and death.

### 2.8. Determination of Antioxidant Activity

#### 2.8.1. DPPH Radical-Scavenging Activity

A DPPH-free radical scavenging assay was used to determine the antioxidant capacity of *Caralluma edulis* extracts as described by [25]. Serial dilutions (25, 50, 100, 200, and 500 μg/mL) of the extract were prepared. One mL of DPPH solution (0.004% in ethanol) was mixed with each dilution. The mixture was then incubated at 37 °C for 30 min. UV–visible spectrophotometer (HC-B030C, Happy care, Guangzhou, China) was set at 517 nm, and absorbance of the samples was measured. The negative control containing absolute ethanol was used for this experiment, whereas the positive control contained ascorbic acid. The following equation calculated the DPPH scavenging activity (%).
DPP scavenging activity (%) = A_0_ − A/A_0_ × 100(3)

A_0_ is the absorbance of the negative control (0.004% DPPH solution), and A is the absorbance of the sample containing the extract. The percentage of residual DPPH against the sample concentration was plotted, and the required concentration that inhibits 50% of DPPH (IC_50_) was estimated.

#### 2.8.2. Ferric Reducing Antioxidant Power (FRAP) Assay

The antioxidant capacity of different extracts of *Caralluma edulis* was also estimated by ferric-reducing ability using a FRAP assay according to the method of Benzie and Strain [26]. The FRAP reagent consisted of a mixture of acetate buffer (300 mM, pH 3.6), 20 mM FeCl_3_, and a solution of 10 mM TPTZ (2,4,6-tri(2-pyridyl)-1,3,5-triazine) in 40 mM HCl at 10:1: (*v*/*v*/*v*). The sample solution (30 μL) was mixed with FRAP reagent (270 μL) and mixture was incubated in the dark at room temperature for 30 min. The absorbance of the samples were measured by UV–visible spectrophotometer (HC-B030C, Happy care, Guangzhou, China) at 595 nm. The reducing power (% inhibition) was measured by the following equation.
% inhibition = A_sample_ − A_control_ /A_sample_ × 100(4)

A_sample_ is absorbance of the test sample and A_control_ is absorbance of blank control (containing all reagents except the extract solution).

### 2.9. Determination of In Vitro Anti-Inflammatory Activity

The inhibitory actions of different extracts of *Caralluma edulis* on albumin denaturation, proteinase activity, and membrane stabilization were determined [27,28] and in vitro anti-inflammatory potential was estimated by these values.

A freeze-dried extract sample of each solvent was used to prepare the serial dilutions (25 to 500 μg/mL) using dimethyl sulfoxide (DMSO) as diluting solvent. The negative and positive controls were aspirin (100 μg/mL, Sigma-Aldrich, Ascent, Singapore) and DMSO, respectively.

#### 2.9.1. Inhibitory Action on Albumin Denaturation

In total, 1 mL of sample was mixed with 1 mL of bovine albumin fraction (1% aqueous solution). The buffer solution was used to adjust the pH of the mixture to 6.3. The reaction mixture was then incubated at 37 °C for 20 min and then heated to 51 °C for 30 min. The mixture was then allowed to cool to room temperature. The absorbance of the mixture was measured at 660 nm with a spectrophotometer (HC-B030C UV-Visible, Happy care, Guangzhou, China). The following formula calculated the percentage of protein denaturation. The IC_50_ values of each extract were determined.
Percentage inhibition (%) = A_control_ − A_sample_/A_control_ × 100(5)
where A_control_ and A_sample_ represent the absorbance of negative control (DMSO) and the tested sample, respectively.

#### 2.9.2. Determination of Antiproteinase Activity

The reaction mixture containing 1 mL of sample, 1 mL of 20 mM Tris HCl buffer (pH 7.4), and 0.06 mg trypsin was incubated at 37 °C for 5 min. A total of 1 mL of 0.7% (*w*/*v*) casein was added to the reaction mixture and again incubated for 20 min. Then 2 mL of 70% perchloric acid (HClO_4_) was gradually added to the mixture and centrifuged (6000 rpm) at 4 °C for 10 min. After centrifugation, the supernatant was separated. A spectrophotometer (HC-B030C UV-Visible, Happy care, Guangzhou, China) was set at 210 nm, and the absorbance of the mixture was measured. The percentage inhibition of proteinase was calculated by Equation (5). The results were expressed as IC_50_ values.

#### 2.9.3. Membrane Stabilization

A healthy human volunteer who had not administered any medicine for two weeks was selected. A blood sample was obtained and centrifuged at 3000 rpm per min for 10 min. The sample was mixed with normal saline, and 10% RBC suspension was prepared. One mL of sample was added to this suspension and incubated at 56 °C for 30 min to observe the effect on heat-induced hemolysis. After incubation, the mixture was cooled and centrifuged at 2500 rpm for 5 min. The absorbance of the supernatant was then measured at 560 nm using a spectrophotometer (HC-B030C UV-Visible, Happy care, Guangdong, China). The percentage inhibition of hemolysis was calculated by Equation (5). The results were expressed as IC_50_ values.

### 2.10. Determination of Antidiabetic Activity

#### 2.10.1. Induction of Diabetes

Alloxan monohydrate (Sigma Chemicals, St. Louis, MO, USA) was mixed in normal saline (5%, *w*/*v*) and administered as a single intravenous injection (marginal ear vein) at a dose of 120 mg/kg, body weight. Blood samples were collected after 5 days, and glucose levels were monitored. The rabbits with blood glucose levels >250 mg/dL were declared diabetic and were used in the experiment.

#### 2.10.2. Acute Effect of the *Caralluma edulis* Extracts in Alloxan-Induced Diabetic Rabbits

A total of 36 rabbits were divided into six groups. Each group had six animals. Group-I was the control group and contained healthy rabbits. They were offered only vehicle (tween 80 in distilled water, 10% (*v*/*v*)) orally in a volume of 10 mL/kg. The other groups (group-II to group-VI) comprised diabetic rabbits. Group-II was offered a reference antidiabetic drug (Glibenclamide at a dose of 5 mg/kg, P.O.) suspended in a vehicle (10 mL/kg). Group III, IV, V, and VI animals received *Caralluma edulis* extracts (water, methanol, ethanol, and acetone) suspended in a vehicle (10 mL/kg). One of the marginal ear veins was exposed, and blood was collected from it before and at 1, 2, and 6 h after dosing in fasting animals to determine the glucose and insulin levels.

#### 2.10.3. Subacute Effect of the *Caralluma edulis* Extracts in Alloxan-Induced Diabetic Rabbits

The *Caralluma edulis* extracts were offered for 8 days to the rabbits, and their subacute antidiabetic effect was observed. Rabbits were divided similar to the division for the acute effect. Blood samples were obtained from one of the marginal ear veins on specified days (1st, 3rd, 5th, and 8th) after dosing, and levels of glucose, insulin, antioxidant enzymes (superoxide dismutase and catalase), and lipid peroxidation were measured. Each blood sample was collected in fasting animals. The effect of different extracts on body weights was also observed on these days.

### 2.11. Analytical Method

#### 2.11.1. Measurement of Blood Glucose Level

Blood glucose level was measured by the commercially available glucose kit (Sigma-Aldrich, Ascent, Singapore) in mg/dL.

#### 2.11.2. Measurement of Insulin Level

Serum insulin level was estimated using the commercially available kit (Sigma-Aldrich, Ascent, Singapore) and was expressed as IU/mL.

#### 2.11.3. Measurement of the Lipid Peroxidation (LPO) in Serum

Lipid peroxidation is commonly estimated by observing the level of thiobarbituric acid reactive substances (TBARS) and malondialdehyde (MDA). The MDA level was determined in serum according to the method of Yoshoiko et al. [29]. Values were presented in nmol/mL.

#### 2.11.4. Estimation of Antioxidant Enzymes

The activity of superoxide dismutase (SOD) was determined in serum according to the method of Sun et al. [30]. The amount of protein that causes the 50% inhibition of NBT rate is defined as one unit of SOD. The activity of catalase (CAT) was estimated in serum according to the method of Yasmineh et al. [31] and expressed as kU/L.

### 2.12. Histopathological Analysis

The animals were sacrificed on the 8th day of administration of plant extracts. Liver specimens were washed with normal saline and fixed with 10% formalin. Fixed tissues were embedded in paraffin wax and sectioned in rotary microtome (5 μm thick). Tissue samples were then stained with hematoxylin and eosin dyes and examined for any histological changes using a light microscope.

### 2.13. Statistical Analysis

All analyses were performed in triplicate; these values were then shown as mean values and their standard derivations (±SD). Analysis of variance (ANOVA) was used, followed by Tukey’s test for multiple comparison tests. *p* values < 0.05 were considered as significant. All the statistical comparisons were analyzed by SPSS software (Version no. 22, IBM, Armonk, NY, USA).

## 3. Results

### 3.1. GC-MS Profiling of Methanolic Extract of Caralluma edulis

A total of 32 compounds were identified from the GC-MS analysis of *Caralluma edulis* plant extract indicating different phytochemical activities. The chromatogram is shown in Figure 1.

The chemical compounds with their molecular formula, retention time, and peak area percentage are presented in Table 1.

### 3.2. Effects of Different Solvents on Extraction Yield

The effect of different solvents (distilled water, methanol, ethanol, and acetone) on the extraction yield of *Caralluma edulis* was examined. The extraction yield varied with these solvents (Figure 2). The results showed that methanol yielded the maximum extraction yield (38.3%), followed by distilled water (26.8%), ethanol (20.3%), and acetone (16.7%).

The impact of these solvents on the chemical composition of *Caralluma edulis* was also determined. Data in Table 2 demonstrated the highest extraction of phenolics (14.3 mg GAE/g DW), flavonoids (1.88 mg QE/g DW), alkaloids (1.44 mg AE/g DW), and terpenoids (1.22%, *w*/*w*) contents in methanolic extract and proved to be the optimal solvent for the extraction of bioactive components from *Caralluma edulis*.

The extraction yield of phenolic compounds (6.20 mg GAE/g DW) was high in distilled water. However, the lowest contents of alkaloids, flavonoids, and terpenoids were extracted in distilled water compared to other solvents. Ethanol also effectively extracted high contents of alkaloids (1.32 mg AE/g DW) and terpenoids (0.97%, *w*/*w*), but less than methanol, and low levels of phenolic (4.44 mg GAE/g DW) and flavonoids contents (0.62 mg QE/g DW).

The data in Table 3 show that all of the tested extracts of *Caralluma edulis* possessed some amount of the investigated phytochemicals.

### 3.3. Impact of Extraction Solvents on Antioxidants and In Vitro Anti-Inflammatory Capacities of Caralluma edulis

#### 3.3.1. Antioxidant Activity of *Caralluma edulis* Extracts

The antioxidant capacities of *Caralluma edulis* extracts were indexed by the radical scavenging activity on the DPPH assay. As shown in Figure 3, the free radical scavenging activities of different extracts were significantly different.

Among the extracts tested, the methanolic extract exhibited strong scavenging activities as expressed by a low IC_50_ value of 17.3 μg/mL. Ethanolic and acetone extracts showed weak scavenging properties by respective IC_50_ values of 29.5 μg/mL and 68.4 μg/mL compared to methanol extract. The water extract possessed the most inadequate scavenging activity among all the tested extracts, with an IC_50_ value of 92.5 μg/mL.

The antioxidant capacity of different extracts from *Caralluma edulis* were also estimated by the FRAP assay that depends upon the reduction of ferric tripyridyltriazine (Fe(III)-TPTZ) complex to the ferrous tripyridyltriazine (Fe(II)-TPTZ) at low pH. It is clear from Table 4 that the methanol extract showed the highest reducing ability (543.8 ± 7.68 μmol Fe(II)/g DW) followed by the ethanol (429.4 ± 9.12 μmol Fe(II)/g DW), acetone (376.1 ± 6.97 μmol Fe(II)/g DW), and distilled water (256.3 ± 8.54 μmol Fe(II)/g DW) extracts.

#### 3.3.2. Anti-Inflammatory Activity of *Caralluma edulis* Extracts

##### Inhibition of Albumin Denaturation

Protein denaturation occurs when proteins lose their biological functions due to the change in their structures. This happens due to other compounds, heat, or any other external stress. Therefore, the denaturation of tissue proteins is considered one of the markers of inflammation. We assessed the in vitro anti-inflammatory activities of *Caralluma edulis* extracts for their inhibitory actions on protein denaturation. Data for the inhibitory activities of *Caralluma edulis* extracts are shown in Table 5.

It is clear from the data in Table 5 that the methanolic extract of the plant showed the most potent inhibitory activity with an IC_50_ value of 26.1 μg/mL, followed by the ethanolic extract (IC_50_ = 36.0 μg/mL) and acetone extract (IC_50_ = 94.3 μg/mL). Aspirin, a standard anti-inflammatory drug, only inhibited 32.4% of albumin denaturation at a 100 μg/mL concentration. Hence, these extracts (methanolic, ethanolic, and acetone) displayed good protein protection compared to aspirin. However, the inhibitory action of the water extract on protein denaturation was very low (IC_50_ = 759 μg/mL).

##### Determination of Antiproteinase Activity

Leukocytes and resident tissue cells produce various proteinases that contribute to the inflammatory process. Thus, the inhibition of proteinases may reduce inflammation. In this study, the antiproteinase activities of different extracts from *Caralluma edulis* were determined, and the results are shown in Table 3. The methanolic extract showed the highest antiproteinase activity (IC_50_ = 408.4 μg/mL), followed by the ethanolic (IC_50_ = 421 μg/mL), acetone (IC_50_ = 576.3 μg/mL), and water extracts (IC_50_ = 589.9 μg/mL). The antiproteinase activities of all these extracts were lower than aspirin, which inhibited only 35.6% of proteinases at a concentration of 100 μg/mL.

##### Determination of Membrane Stabilization Effect

The membranes of RBCs were ruptured by heat, and the membrane stabilization abilities of different extracts from *Caralluma edulis* were observed in this study. The data in Table 3 reveal that the highest protection against heat-induced hemolysis was observed in methanolic extract (IC_50_ = 319.3 μg/mL). The ethanolic extract also provided good protection to the RBCs’ membrane against heat-induced damage. However, the water and acetone extracts showed low protective activity compared to the other extracts. Aspirin showed 28.2% protection at a concentration of 100 μg/mL compared to the *Caralluma edulis* extracts. These results suggest that *Caralluma edulis* extracts may provide protection to the cell membrane, hence, suppressing the inflammatory reaction.

### 3.4. Acute Toxicity Studies

Rabbits with high doses (250 mg/kg) remained alive for 14 days and did not show any signs of acute toxicity.

### 3.5. Determination of Antidiabetic Activity

#### 3.5.1. Acute Effect of the *Caralluma edulis* Extracts in Alloxan-Induced Diabetic Rabbits

The acute antidiabetic effect of *Caralluma edulis* in different solvents was tested in alloxan-induced diabetic rabbits. As shown in Table 6, none of the tested extracts of *Caralluma edulis* showed any remarkable difference in blood glucose levels during the observed hours (0 to 6 h). Glibenclamide showed marked antidiabetic activity at 1 to 6 h of treatment in the diabetic rabbits.

Table 7 indicates the effect of *Caralluma edulis* extracts on the serum insulin level in diabetic rabbits. All of the tested extracts of *Caralluma edulis* showed no notable variation in blood insulin levels during the observed hours (0 to 6 h). Glibenclamide showed a remarkable increase in the insulin level at 1 hour (19.80%) and at 6 h (34.5%) of treatment in the diabetic rabbits.

#### 3.5.2. Subacute Effect of the *Caralluma edulis* Extracts in Alloxan-Induced Diabetic Rabbits

Different extracts of *Caralluma edulis* were offered to the alloxan-induced rabbits for eight consecutive days to investigate their subacute antidiabetic effects. Each animal’s blood glucose and insulin levels were monitored on the 1st, 3rd, 5th, and 8th days post-administration. The data in Table 8 clearly demonstrate the high level of the glucose levels in the diabetic control rabbits compared with those in the control rabbits throughout the experiment.

On day one, the methanolic extract of *Caralluma edulis* significantly decreased the blood glucose level as compared to the diabetic control. On the 3rd, 5th and 8th days, all of the tested plant extracts of *Caralluma edulis* demonstrated a significant decrease in the blood glucose levels compared with those of the diabetic control rabbits; however, the decreasing effect was more pronounced in the diabetic rabbits treated with the methanolic plant extract during this period. Glibenclamide also caused a significant drop in the blood glucose levels in a similar pattern as shown by the methanolic plant extract.

The antidiabetic effect of *Caralluma edulis* plant extracts on the serum insulin levels is shown in Table 9.

On day one, the *Caralluma edulis* methanolic extract showed a significant increase in the insulin levels, and this effect was more pronounced during the 3rd day up to the 8th day of the experiment. The ethanolic, acetone, and water extracts of *Caralluma edulis* did not show any change in the insulin levels on the 1st day. Nevertheless, these extracts demonstrated a substantial increase in the insulin levels from the 3rd day to the 8th day of the experiment. Glibenclamide also showed a rise in the insulin levels similar to those of the methanolic extract of *Caralluma edulis*.

#### 3.5.3. Determination of Antioxidant Enzymes

In vivo antioxidant effects of the different extracts from *Caralluma edulis* were investigated by monitoring the serum concentrations of SOD, MDA, and CAT on the 1st, 3rd, 5th, and 8th days post administration. The data in Table 10 revealed the high level of lipid peroxidation in the diabetic rabbits, as evidenced by the high values of MDA compared with the control group.

All of the extracts of *Caralluma edulis* importantly suppressed the increase in the MDA level on the 5th and 8th day of the experiment. In addition, Glibenclamide also reduces lipid peroxidation in serum by inhibiting the increase in MDA on these days.

The activities of the antioxidant enzymes (SOD and CAT) of all groups are presented in Table 8. There was a significant decrease in the SOD and CAT levels in the diabetic rabbits compared with those in healthy normal rabbits. The diabetic rabbits treated with the different extracts from *Caralluma edulis* did not show any significant change in the antioxidant enzymes during the 1st day and the 3rd day; however, the actions of SOD and CAT were remarkably improved on the 5th and 8th days in all of the treated groups.

### 3.6. Effect of Different Extracts from Caralluma edulis on Growth Rate of Experimental Animals

As shown in Table 11, all of the treated plant extracts positively affected the health of the diabetic rabbits. A gradual increase in weight was observed in all of the treated groups.

### 3.7. Histopathological Examination

The histopathological examinations of the tested extracts are shown in Figure 4. The tested dose of 200 mg/kg in all of the tested solvents was safe, and no histopathological changes were observed in the liver samples of the experimental animals of different groups.

## 4. Discussion

The *Caralluma edulis* plant is used traditionally as a vegetable against certain diseases by the local inhabitants. Bioactive compounds present in the plant play an important role in maintaining health, and these can be obtained by the extraction process [32]. Multiple factors such as selection of the solvent, the chemistry of phytoconstituents, extraction technique, temperature, and time are strongly associated with the efficiency of the extraction procedure [33]. Among these factors, the selection of a particular solvent plays a vital role in the extraction method.

We have studied the impact of different solvents (distilled water, methanol, ethanol, and acetone) on the extraction yield and phytochemical contents. The extraction yield varies with these different solvents. The solvents tested vary in their polarity with each other, which could be the reason for different extraction yields. Our results showed a greater higher extraction yield in more polar solvents (methanol and distilled water) than acetone, which is less polar. This result is consistent with some previous studies [34,35]. It also indicates that *Caralluma edulis* contains high polar compounds soluble in highly polar solvents (methanol, distilled water). However, further studies should be conducted to better understand the impact of solvents on the composition of phytoconstituents during the extraction process.

The contents of phytoconstituents (alkaloids, phenolics, terpenoids, and flavonoids) in the extraction yield varied in different solvents. The methanolic extract resulted in the highest levels of these compounds, indicating these bioactive compounds’ higher solubility than the other tested solvents. Based on these findings, it can be suggested that methanol is the optimal solvent for extracting phytoconstituents from the *Caralluma edulis* plant.

Since the extraction solvents affect the composition of the phytochemicals and their yield during the extraction process, hence, it could be assumed that they affect the pharmacological action of the extract [36,37]. In our study, the antioxidant potential of different extracts from *Caralluma edulis* was observed via DPPH scavenging activity and FRAP reducing power assays. The IC_50_ values of both assays suggest that the methanolic extract possesses the strongest scavenging activity among all tested solvents. This is possibly due to it having the highest level of bioactive phytoconstituents [38,39]. These phytochemicals have strong defensive potential against the harmful effects of oxidative damage by scavenging diverse ROS, including superoxide anions, hydroxyl radicals, hypochlorous acid, peroxyl radicals, and peroxynitrite [40]. These observations suggest that the methanolic extract of *Caralluma edulis* could be a potential antioxidant candidate for drug development.

Inflammation is a complex biological response that depends on multiple factors. One of the main factors is the denaturation of tissue protein. Hence, we have studied the inhibitory action of different extracts from *Caralluma edulis* to assess the in vitro anti-inflammatory activity of the plant. The results showed that the methanolic, ethanolic, and acetone extracts of *Caralluma edulis* strongly inhibited protein denaturation. This protective capacity was higher than aspirin, suggesting that *Caralluma edulis* extracts could be potential candidates as anti-inflammatory agents.

Since proteinases are usually involved in the inflammatory process, we also studied the antiproteinase activities of different extracts of the plant. Results showed that *Caralluma edulis* extracts have antiproteinase activity. The RBC membrane stabilization potential is the index of anti-inflammatory activity. The membrane structures of RBC and lysosomes are similar. During the inflammatory reaction, the lysosomes are lysed, and certain compounds have been released that result in inflammatory reactions and other complications. Membrane stabilization is a biological process that solidifies the lysosomal membrane against impairments. It prevents the release of lysosomal components made by inflammatory mediators and minimizes the inflammatory process [4]. In our study, the methanolic extract of the plant exhibited the highest protection to the RBC membrane against heat-induced damage, suggesting that the extract has great potential to stabilize lysosomal membranes. This activity might be due to bioactive compounds that can interfere with phospholipases and protect lysosomal membranes against injury. The results suggest *Caralluma edulis* extracts possess anti-inflammatory potential, as evidenced by the stabilization effect of the RBC membrane against heat.

The inhibitory activities of protein denaturation, proteinases, and membrane stabilization activities suggest the anti-inflammatory potential of *Caralluma edulis* extracts that might be because of the phytoconstituents (phenolics, flavonoids, terpenoids, and alkaloids) present in the plant. Recent investigations have shown the strong anti-inflammatory potential of these compounds [41,42]. Taken together, it may be assumed that the methanolic extract of *Caralluma edulis* is the most promising source of antioxidant and anti-inflammatory activities.

The traditional use of *Caralluma edulis* in various medical disorders provoked us to conduct the experimental study for its scientific validation. There are a few previous reports regarding the antihyperlipidemic effect of *Caralluma edulis* in high-fat-diet-induced hyperlipidemia in rats [43]. This study tested different extracts from *Caralluma edulis* for their action in acute and subacute diabetic rabbit models. The methanolic plant extract of *Caralluma edulis* significantly lowered the glucose level. It remarkably increased the insulin level in the acute diabetic rabbit model, and these effects were almost the same as those of synthetic drug Glibenclamide.

The other extracts of *Caralluma edulis* did not show any antidiabetic activity in this model. On the other hand, all of the extracts from *Caralluma edulis* showed antidiabetic potential by significantly decreasing the blood glucose level and increasing the insulin level in the subacute diabetic rabbit model. However, the methanolic extract of *Caralluma edulis* demonstrated the parallel antidiabetic activity as shown by the synthetic drug Glibenclamide in this scenario.

Alloxan induces chemical diabetes by damaging the insulin-secreting cells of the pancreas. This damage involves many cells and leads to a lesser amount of insulin release, which paves the way for the decreased utilization of glucose by the tissue [44]. Glibenclamide is a member of the sulphonylurea class that decreases the blood glucose level by the release of insulin from the pancreatic cells. The possible underlying mechanism for the antidiabetic activity of the *Caralluma edulis* extracts might be correlated to the presence of such phytochemicals that can induce insulin secretion from the existing residual cell of islets.

The subacute antihyperglycemic and insulinotropic effects of the *Caralluma edulis* extracts (200 mg/kg) might be due to potent phytochemicals that have a significant role in decreasing the blood glucose level and enhancing the insulin levels. In this study, all tested extracts have a considerable amount of critical bioactive compounds (phenolic, flavonoids, alkaloids, and terpenoids). Recent studies provide strong evidence of these bioactive compounds in managing chemical-induced diabetes in experimental animals [45,46]. Different alkaloids exert their action by various mechanisms. Berberine is supposed to inhibit alpha-glucosidase activity, thus decreasing the glucose transport through the intestinal epithelium [47].

Similarly, casuarine 6-*O*-α-glucoside also inhibits the α-glucosidase activity [15]. Tecomine is an alkaloid and can increase the adipocyte glucose uptake rate [48]. Some other alkaloids, such as catharanthine, vindolinine, vindoline, and leurosine are supposed to reduce the blood glucose levels in normal and alloxan diabetic rabbits [49]. Extracts of *Caralluma edulis* are enriched in flavonoids that also have antidiabetic properties. Anthocyanins are potent flavonoids and should have a hypoglycemic effect by decreasing the intestinal absorption of glucose [50]. Epigallocatechin gallate, a vital flavonoid, also has glucose-lowering properties by reducing glucose production and enhancing the tyrosine phosphorylation of the insulin receptor and insulin receptor substrate-1 (IRS-1)-like insulin. Additionally, this compound increases the activities of mitogen-activated protein kinase, phosphoinositide 3-kinase, and p70 (s6k) and subsequently mimics the insulin [51].

GC-MS analysis demonstrated that *Caralluma edulis* is enriched in some phytocompounds that are known for their antidiabetic potentials. Previous studies have revealed that some phytosterols, including β-Sitosterol, have a reducing effect on blood glucose levels, enhance the antioxidant status of pancreatic cells, and promote serum insulin production [52]. Additionally, their antidiabetic action may also be attributed to the inhibitory effect on glycolytic enzymes such as α-amylase. β-Sitosterol was found to inhibit α-amylase enzyme as an uncompetitive inhibitor [53].

*Caralluma edulis* extracts might have some insulin-tropic substances that protect the functional cells from alloxan-induced damage or regenerate the damaged cells for proper functioning. *Caralluma edulis* is used as a folk medicine in diabetic patients. In a previous study, *Caralluma edulis* was tested for its antidiabetic potential at doses of 2 mg/kg to 4 mg/kg in alloxan-induced diabetic rabbits and did not observe any antidiabetic activity at these tested doses [54]. In the present study, different extracts from *Caralluma edulis* were tested at a 200 mg/kg dose and found to have antidiabetic potential.

Oxidative stress may be correlated with pathological changes in diabetic animals [55]. The high blood glucose level may generate it for a long period. Persistent hyperglycemia weakens the antioxidative defense system and encourages de novo free radicals’ generation [56]. Oxygen-free radicals react with different cell membrane constituents, especially polyunsaturated fatty acids. These reactions result in lipid peroxidation [57]. Increased lipid peroxidation disturbs the membrane fluidity and impairs the membrane-bound enzymes and receptors [58]. Biological compounds with antioxidant potential may improve this cellular damage [59]. The MDA level usually represents the LPO status. In the current study, the higher levels of MDA in the treated groups than the control group indicate the production of free radicals. All of the tested *Caralluma edulis* extracts in alloxan diabetic rabbits in subacute diabetic experiments demonstrated a significant decrease in MDA levels. The reduced MDA level might enhance the glutathione peroxidase (GPX) activity in rabbits treated with *Caralluma edulis* extracts, hindering the LPO reactions [60].

Enzymatic antioxidants (SOD and CAT) play a crucial role in directly eliminating ROS [61]. These two enzymes provide cellular protection from the damage of free radicals. Persistent hyperglycemia favors free radical generation, and both of these enzymes are diminished in diabetic conditions due to non-enzymatic oxidation and glycosylation [62]. The results of our study are consistent with some previous studies that reported the reduced activities of SOD and CAT in diabetic animals [63,64]. The possible reason might be alloxan-induced ROS generation that limits their actions. Long-term treatment of diabetes with high doses of *Caralluma edulis* extracts reversed the SOD and CAT functions, possibly due to decreasing oxidative stress as evidenced by a low level of lipid peroxidation. Recent studies suggest that plants enriched with polyphenolic compounds inhibit LPO and promote the activity of glutathione peroxidase, which provides cellular protection by neutralizing ROS [65]. The presence of several bioactive antioxidant principles in *Caralluma edulis* extracts may provide improved antioxidant status in diabetic rabbits.

## 5. Conclusions

We studied the impact of different solvents on the extraction of bioactive compounds from the *Caralluma edulis* plant. The methanolic extract yielded the highest extraction yield and the highest flavonoid, phenolic, alkaloid, and terpenoid contents. It seems to be the optimal solvent for extracting the bioactive compounds from *Caralluma edulis*. The methanol extract demonstrated the highest anti-inflammatory activity and antioxidant potential among the extracts tested. These findings suggest that the methanol extract is a potent anti-inflammatory and antioxidant candidate for the pharmaceutical and nutraceutical industries.

The antidiabetic activities of different extracts from *Caralluma edulis* in acute and subacute diabetic models were also studied. The tested extracts did not show any significant antidiabetic effect in the acute diabetic model; however, these extracts exhibited antidiabetic potential at their tested doses (200 mg/kg) in the subacute diabetic model. They can inhibit lipid peroxidation and activate the antioxidant enzymes (CAT and SOD) in diabetes. The observed antioxidant potential of these extracts may be partially responsible for their antidiabetogenic properties. Our study provides scientific evidence for the folk use of the *Caralluma edulis* plant against diabetes. However, further studies should be conducted to explore the detailed mechanism of action of bioactive compounds of *Caralluma edulis* in diabetes.

## Figures and Tables

**Figure 1 molecules-27-05346-f001:**
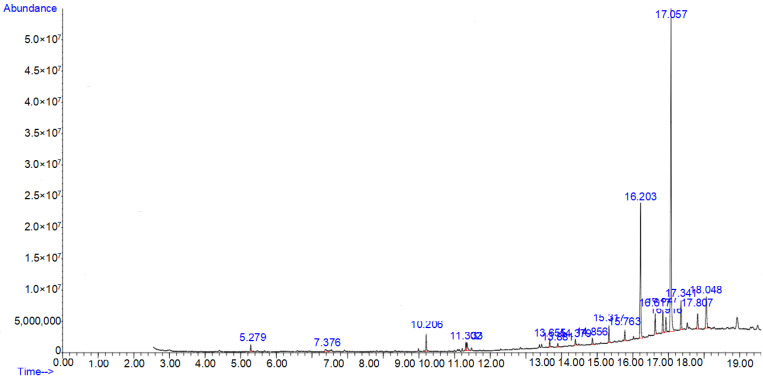
GC-MS chromatogram of methanolic extract of *Caralluma edulis* plant.

**Figure 2 molecules-27-05346-f002:**
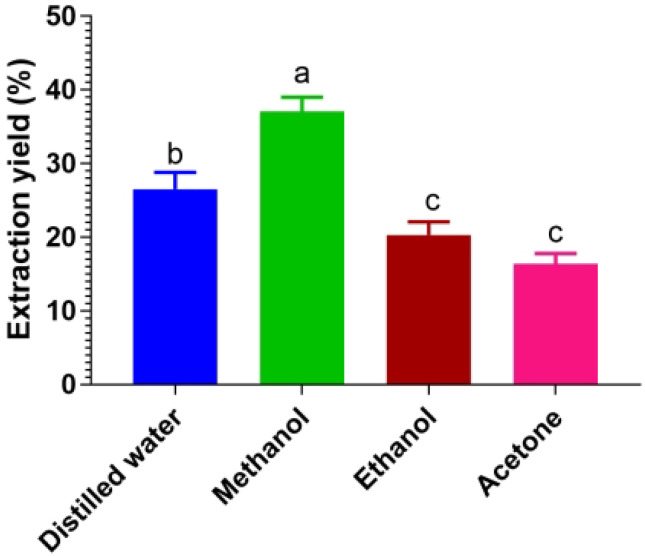
Extraction yield of *Caralluma edulis* using different solvents. All values are the mean ± SD (*n =* 3). Means of a column with different letters significantly differ by Tukey’s test at *p* < 0.01.

**Figure 3 molecules-27-05346-f003:**
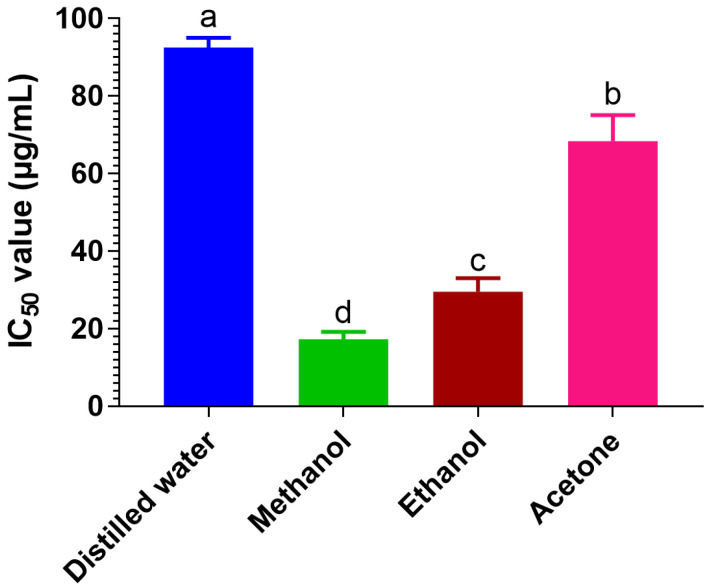
The 50% inhibitory concentration (IC_50_) values of DPPH scavenging activity of different *Caralluma edulis* extracts. All values are the mean ± SD (*n =* 3). Means of a column with different letters significantly differ by Tukey’s test at *p* < 0.05.

**Figure 4 molecules-27-05346-f004:**
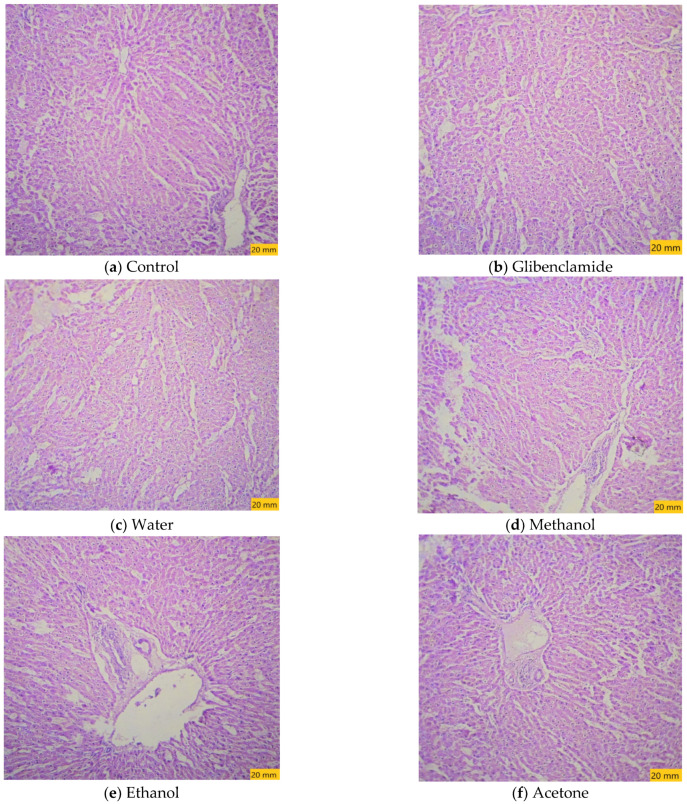
Photographs of the liver tissues from different groups (20×, H&E): (**a**) control; (**b**) Glibenclamide (5 mg/kg) reference drug groups; (**c**) *Caralluma edulis* water extract (200 mg/kg); (**d**) *Caralluma edulis* methanol extract (200 mg/kg); (**e**) *Caralluma edulis* ethanol extract (200 mg/kg); (**f**) *Caralluma edulis* acetone extract (200 mg/kg).

**Table 1 molecules-27-05346-t001:** Phytoconstituents in the methanol extract of *Caralluma edulis* plant as detected by GC-MS analysis.

S. No	RT in min	Name of the Compound	Molecular Formula	Peak Area %
1	5.279	Propane, 1,2,3,-trimethoxy-	C_6_H_14_O_3_	0.91
2	5.279	2-Oxoacetic acid, ethyl ester, oxime	C_4_H_7_NO_3_	0.91
3	7.376	(*E*)-Stilbene	C_14_H_12_	0.94
4	7.376	2,3,5,6-Tetrafluoroanisole	C_7_H_4_F_4_O	0.94
5	7.376	Butyric acid, 4-amino-3-(4-methoxyphenyl)-	C_13_H_19_NO_3_	0.94
6	10.206	*n*-Hexadecanoic acid (Palmitic acid)	C_16_H_32_O_2_	2.09
7	10.206	Pentadecanoic acid	C_15_H_30_O_2_	2.09
8	10.206	Octadecanoic acid	C_18_H_36_O_2_	2.09
9	11.302	9,12-Octadecadienoic acid (*Z,Z*)-	C_18_H_32_O_2_	0.91
10	11.333	*cis*-Vaccenic acid	C_18_H_34_O_2_	1.52
11	11.333	*cis*-13-Octadecenoic acid	C_18_H_34_O_2_	1.52
12	11.333	*trans*-13-Octadecenoic acid	C_18_H_34_O_2_	1.52
13	13.655	Bis(2-ethylhexyl) phthalate	C_24_H_38_O_4_	1.03
14	13.655	Phthalic acid, cyclohexyl 2-pentyl ester	C_19_H_26_O_4_	1.03
15	13.881	Heptadecane	C_17_H_36_	0.59
16	14.379	Heptacosane, 1-chloro-	C_27_H_55_Cl	0.84
17	14.856	Tetracosane	C_24_H_50_	0.93
18	16.203	Hentriacontane	C_31_H_64_	17.63
19	16.203	Triacontane	C_30_H_62_	17.63
20	16.827	Octadecamethyloctasiloxane	C_18_H_54_O_7_Si_8_	2.88
21	16.827	1,1,3,3,5,5,7,7,9,9,11,11,13,13-Tetradecamethylheptasiloxane	C_14_H_42_O_6_Si_7_	2.88
22	16.827	1*H*-Indole, 1-methyl-2-phenyl-	C_15_H_13_N	2.88
23	16.916	Benzo[*h*]quinoline, 2,4-dimethyl-	C_15_H_13_N	1.92
24	16.916	Cyclotrisiloxane, hexamethyl-	C_6_H_18_O_3_Si_3_	1.92
25	17.057	Tetracosane	C_24_H_50_	49.63
26	17.057	1-Iodohexadecane	C_16_H_33_I	49.63
27	17.057	Octadecane	C_18_H_38_	49.63
28	17.341	gamma-sitosterol	C_29_H_50_O	4.12
29	17.341	beta-sitosterol	C_29_H_50_O	4.12
30	17.807	1-Benzazirene-1-carboxylic acid, 2,2,5a-trimethyl-1a-[3-oxo-1-butenyl] perhydro-, methyl ester	C_15_H_23_NO_3_	2.37
31	17.807	9,10-Methanoanthracen-11-ol,9,10-dihydro-9,10,11-trimethyl-	C_18_H_18_O	2.37
32	18.048	Eicosane	C_20_H_42_	5.63

**Table 2 molecules-27-05346-t002:** Effect of different solvents on phenolic, flavonoid, and alkaloid content of *Caralluma edulis*.

Extraction Solvent	Phenolics (mg GAE/g DW)	Flavonoids (mg QE/g DW)	Alkaloids (mg AE/g DW)	Terpenoids (%, g/g)
Distilled water	6.20 ± 0.99 ^b^	0.63 ± 0.11 ^c^	0.14 ± 0.01 ^d^	0.45 ± 0.05 ^c^
Methanol	14.3 ± 1.95 ^a^	1.88 ± 0.18 ^a^	1.44 ± 0.22 ^a^	1.22 ± 0.09 ^a^
Ethanol	4.44 ± 0.60 ^bc^	0.62 ± 0.13 ^c^	1.32 ± 0.17 ^ab^	0.97 ± 0.14 ^ab^
Acetone	2.91± 0.75 ^c^	0.76 ± 0.12 ^bc^	0.88 ± 0.19 ^bc^	0.52 ± 0.07 ^c^

All values are the mean ± SD (*n* = 3). Means within a column with different letters significantly differ by Tukey’s test at *p* < 0.05.

**Table 3 molecules-27-05346-t003:** Phytochemicals’ concentration of different extracts from *Caralluma edulis* measured by spectrophotometer.

Name of Compound and Class	Wavelength	Pharmacological Activities	*C. edulis* (Water ext.)	*C. edulis* (Methanol ext.)	*C. edulis* (Ethanol ext.)	*C. edulis* (Acetone ext.)
Campesterol (Phytosterol)	254 nm	Antioxidant, anti-inflammatory, antidiabetic	0.309	0.205	0.368	0.265
β-sitosterol (Phytosterol)	210 nm	Antioxidant, anti-inflammatory, antidiabetic	0.159	0.324	0.279	1.239
(*E*)-Stilbene (Phenolic compound)	285 nm	Antioxidant, hepatoprotective, anti-inflammatory	0.347	0.651	0.418	0.517
Gallic acid (Phenolic compound)	256 nm	Radical scavenging, anti-inflammatory, immunoregulatory	0.884	1.341	0.947	0.748
Pentadecanoic acid (Fatty acid)	364 nm	Antidiabetic	0.336	0.427	0.254	0.387

**Table 4 molecules-27-05346-t004:** FRAP assay activity and IC_50_ values of different extracts from *Caralluma edulis* plant.

Extraction Solvent	FRAP (μmol Fe (II)/g DW)	IC_50_ (μg/mL)
Distilled water	256.3 ± 8.54 ^d^	369.2 ± 9.11 ^d^
Methanol	543.8 ± 7.68 ^a^	175.5 ± 3.67 ^a^
Ethanol	429.4 ± 9.12 ^b^	230.6 ± 4.68 ^b^
Acetone	376.1± 6.97 ^c^	282.7 ± 5.53 ^c^

All values are the mean ± SD (*n* = 3). Means within a column with different letters significantly differ by Tukey’s test at *p* < 0.05.

**Table 5 molecules-27-05346-t005:** In vitro anti-inflammatory activity of different *Caralluma edulis* extracts.

Samples	IC_50_ Values (µg/mL)		
	Albumin Denaturation	Proteinase Inhibition	Membrane Stabilization
Water extract	759.0 ± 19.9 ^a^	589.9 ± 22.5 ^a^	835.1 ± 21.7 ^a^
Methanolic extract	26.1 ± 2.25 ^c^	408.4 ± 15.3 ^c^	319.3 ± 13.6 ^cd^
Ethanolic extract	36.0 ± 4.33 ^c^	421.0 ± 12.6 ^c^	355.8 ± 18.7 ^c^
Acetone extract	94.3 ± 6.97 ^b^	576.3 ± 17.2 ^ab^	586.4 ± 19.1 ^b^
		Inhibitory activity, %	
Aspirin (100 µg/mL)	32.4 ± 5.4	35.6 ± 2.3	28.2 ± 1.9

All values are the mean ± SD (*n* = 3). Means within a column with different letters significantly differ by Tukey’s test at *p* < 0.05.

**Table 6 molecules-27-05346-t006:** Acute effect of *Caralluma edulis* extracts on blood glucose levels in alloxan-induced diabetic rabbits.

Group	Dose (mg/kg)	Mean Blood Glucose Concentration + S.D. (mg/dL)			
		0 h	1 h	3 h	6 h
Control (healthy)	10	117 ± 7.02	128 ± 5.03	126 ± 2.08	117 ± 4.16
Glibenclamide	5	121 ± 9.50	106 ± 4.16 *	92 ± 5.27 **	82 ± 8.93 **
*Caralluma edulis* (water extract)	200	114 ± 5.29	124 ± 6.50	114 ± 9.01	109 ± 3.05
*Caralluma edulis* (methanol extract)	200	109 ± 7.09	119 ± 7.02	115 ± 3.51	104 ± 2.43
*Caralluma edulis* (ethanol extract)	200	118 ± 7.02	111 ± 11.06	118 ± 3.67	107 ± 3.46
*Caralluma edulis* (acetone extract)	200	115 ± 6.52	112 ± 1.70	117 ± 2.56	108 ± 3.65

* *p* < 0.05 significant from the control animals, ** *p* < 0.01 significant from the control animals.

**Table 7 molecules-27-05346-t007:** Acute effect of *Caralluma edulis* extracts on serum insulin levels in alloxan-induced diabetic rabbits.

Group	Dose (mg/kg)	Mean Serum Insulin Level + S.D. (µIU/dL)			
		0 h	1 h	3 h	6 h
Control (healthy)	10	11.66 ± 1.65	12.17 ± 1.20	13.2 ± 0.60	12.41 ± 0.56
Glibenclamide	5	12.46 ± 2.54	14.58 ± 0.85 *	16.31 ± 0.95 **	18.53 ± 0.71 **
*Caralluma edulis* (water extract)	200	11.76 ± 1.30	13.13 ± 0.38	14.50 ± 0.65	12.05 ± 0.48
*Caralluma edulis* (methanol extract)	200	11.50 ± 0.79	13.66 ± 0.56	14.62 ± 0.60	14.1 ± 0.50
*Caralluma edulis* (ethanol extract)	200	11.93 ± 1.50	13.18 ± 0.94	14.70 ± 0.55	12.40 ± 0.45
*Caralluma edulis* (acetone extract)	200	11.56 ± 0.96	13.12 ± 0.62	14.49 ± 0.54	12.3 ± 0.36

* *p* < 0.05 significant from the control animals, ** *p* < 0.01 significant from the control animals.

**Table 8 molecules-27-05346-t008:** Subacute effect of *Caralluma edulis* extracts on blood glucose levels in alloxan-induced diabetic rabbits.

Group	Dose (mg/kg)	Mean Blood Glucose Concentration + S.D. (mg/dL)			
		1st Day	3rd Day	5th Day	8th Day
Control (healthy)	10	121.6 ± 8.14	113.1 ± 7.54	118.3 ± 7.09	110.6 ± 7.05
Diabetic control ^(a)^	10	427.9 ± 12.01 ***	451.5 ± 12.01 ***	428.4 ± 8.50 ***	374.6 ± 4.04 ***
Glibenclamide ^(b)^	5	399.7 ± 9.60 *	388.8 ± 11.06 **	379.6 ± 9.49 **	342.6 ± 7.37 **
*Caralluma edulis* (water extract) ^(b)^	200	406.2 ± 8.54	414.2 ± 6.02 *	400.1 ± 9.01 *	356.2 ± 3.60 *
*Caralluma edulis* (methanol extract) ^(b)^	200	401.8 ± 6.02 *	394.3 ± 15.01 **	390.2 ± 8.86 **	346.6 ± 5.85 **
*Caralluma edulis* (ethanol extract) ^(b)^	200	410.1 ± 10.59	419.6 ± 10.5 *	399.6 ± 6.35 *	357.1 ± 3.60 *
*Caralluma edulis* (acetone extract) ^(b)^	200	419.6 ± 6.80	417.2 ± 7.02 *	402.3 ± 10.01 *	358.3 ± 6.80 *

* *p* < 0.05 significant from the control animals, ** *p* < 0.01 significant from the control animals, *** *p* < 0.001 significant from the control animals, ^(a)^ compared to vehicle control, ^(b)^ compared to diabetic control.

**Table 9 molecules-27-05346-t009:** Subacute effect of *Caralluma edulis* extracts on serum insulin levels in alloxan-induced diabetic rabbits.

Group	Dose (mg/kg)	Mean Serum Insulin Level + S.D. (µIU/dL)			
		1st Day	3rd Day	5th Day	8th Day
Control (healthy)	10	13.48 ± 0.46	13.25 ± 0.40	13.70 ± 0.14	14.16 ± 0.19
Diabetic control ^(a)^	10	5.95 ± 0.04 ***	5.70 ± 0.12 ***	5.83 ± 0.15 **	6.03 ± 0.71 ***
Glibenclamide ^(b)^	5	6.13 ± 0.08 *	6.17 ± 0.09 **	6.22 ± 0.05 **	6.48 ± 0.26 **
*Caralluma edulis* (water extract) ^(b)^	200	6.08 ± 0.23	6.01 ± 0.28 *	6.12 ± 0.13 *	6.27 ± 0.38 *
*Caralluma edulis* (methanol extract) ^(b)^	200	6.12 ± 0.05 *	6.15 ± 0.07 **	6.18 ± 0.11 **	6.43 ± 0.15 **
*Caralluma edulis* (ethanol extract) ^(b)^	200	6.01 ± 0.09	5.98 ± 0.12 *	6.13 ± 0.21 *	6.25 ± 0.22 *
*Caralluma edulis* (acetone extract) ^(b)^	200	6.04 ± 0.17	5.97 ± 0.15 *	6.12 ± 0.17 *	6.23 ± 0.13 *

* *p* < 0.05 significant from the control animals, ** *p* < 0.01 significant from the control animals, *** *p* < 0.001 significant from the control animals, ^(a)^ compared to vehicle control, ^(b)^ compared to diabetic control.

**Table 10 molecules-27-05346-t010:** Malondialdehyde (MDA), superoxide dismutase (SOD) and catalase levels in serum of alloxan-induced diabetic rabbits after treated with *Caralluma edulis* extracts.

Group	Dose (mg/kg)	Mean Serum SOD (U/mL), MDA (nmol/mL) and Catalase Levels (kU/I) + S.D.			
**Control**	10	**1st Day**	**3rd Day**	**5th Day**	**8th Day**
SOD		87.8 ± 7.53	86.4 ± 8.56	88.6 ± 7.76	87.6 ± 1.85
MDA		4.43 ± 0.47	4.33 ± 0.08	4.63 ± 0.12	4.71 ± 0.17
Catalase		54.2 ± 2.54	58.0 ± 1.47	59.7 ± 1.10	55.2 ± 1.15
**Diabetic control** ^(a)^					
SOD		53.5 ± 6.75 ***	55.4 ± 5.06 ***	60.3 ± 7.05 ***	60.4 ± 1.95 ***
MDA		9.17 ± 0.16 ***	9.31 ± 0.14 ***	9.39 ± 0.14 ***	9.75 ± 0.16 ***
Catalase		31.7 ± 0.66 ***	32.8 ± 1.51 ***	34.6 ± 0.75 ***	39.2 ± 0.26 ***
**Glibenclamide** ^(b)^	5				
SOD		58.1 ± 5.27	63.4 ± 9.09	76.1 ± 2.51 **	72.9 ± 3.09 **
MDA		8.72 ± 0.06	8.84 ± 0.15	8.58 ± 0.17 **	7.77 ± 0.18 **
Catalase		29.2 ± 1.68	34.9 ± 1.07	39.0 ± 0.72 **	48.0 ± 1.41 **
***Caralluma edulis* (water ex.)** ^(b)^	200				
SOD		55.1 ± 6.51	57.9 ± 7.60	72.1 ± 2.98 *	69.7 ± 0.75 **
MDA		8.89 ± 0.23	8.88 ± 0.21	8.83 ± 0.15 *	8.47 ± 0.18 **
Catalase		30.6 ± 1.05	31.5 ± 1.05	37.0 ± 1.25 *	41.4 ± 0.37 *
***Caralluma edulis* (methanol ex.)** ^(b)^	200				
SOD		58.6 ± 5.93	61.5 ± 8.12	75.8 ± 1.51 **	71.4 ± 1.15 **
MDA		8.77 ± 0.12	8.84 ± 0.18	8.65 ± 0.12 **	7.86 ± 0.20 **
Catalase		28.8 ± 0.70	33.0 ± 1.20	38.1 ± 0.55 **	47.7 ± 0.43 **
***Caralluma edulis* (ethanol ex.)** ^(b)^	200				
SOD		59.4 ± 6.70	60.5 ± 8.55	70.7 ± 1.71 *	69.4 ± 1.09 **
MDA		8.78 ± 0.20	8.87 ± 014	7.78 ± 0.14 *	8.62 ± 0.27 **
Catalase		29.4 ± 0.88	30.1 ± 0.57	37.1 ± 1.66 *	42.4 ± 0.66 *
***Caralluma edulis* (acetone ex.)** ^(b)^	200				
SOD		57.9 ± 7.40	58.6 ± 6.57	71.9 ± 1.62 *	70.4 ± 1.05 **
MDA		8.76 ± 0.22	8.84 ± 0.19	8.80 ± 0.25 *	8.50 ± 0.28 **
Catalase		29.1 ± 1.56	30.9 ± 0.43	37.0 ± 0.40 *	41.9 ± 0.52 *

* *p* < 0.05 significant from the control animals, ** *p* < 0.01 significant from the control animals, *** *p* < 0.001 significant from the control animals, ^(a)^ Compared to vehicle control, ^(b)^ Compared to diabetic control.

**Table 11 molecules-27-05346-t011:** Subacute effect of *Caralluma edulis* extracts on body weights in alloxan-induced diabetic rabbits.

Group	Dose (mg/kg)	Mean Body Weight + S.D. (kg) Percent Increase from the Initial Weight			
		1st Day	3rd Day	5th Day	8th Day
Control (healthy)	10	2.19 ± 0.5	2.26 ± 0.4 (3.1%)	2.3 ± 0.3 (5.3%)	2.32 ± 0.6 (9.3%)
Diabetic control ^(a)^	10	2.40 ± 0.8	2.43 ± 0.3 (3.8%)	2.47 ± 0.6 (4.4%)	2.52 ± 0.7 (7.6%)
Glibenclamide ^(b)^	5	2.73 ± 0.7	2.77 ± 0.6 (2.9%)	2.84 ± 0.2 (4.6%)	2.91 ± 0.5 (8.1%)
*Caralluma edulis* (water extract) ^(b)^	200	2.58 ± 0.6	2.63 ± 0.8 (3.1%)	2.67 ± 0.8 (5.1%)	2.73 ± 0.4 (7.0%)
*Caralluma edulis* (methanol extract) ^(b)^	200	2.79 ± 0.7	2.86 ± 0.9 (2.3%)	2.92 ± 0.7 (6.4%)	2.95 ± 0.8 (8.1%)
*Caralluma edulis* (ethanol extract) ^(b)^	200	2.29 ± 0.5	2.32 ± 0.2 (3.3%)	2.41 ± 0.3 (7.5%)	2.49 ± 0.9 (9.5%)
*Caralluma edulis* (acetone extract) ^(b)^	200	2.35 ± 0.4	2.38 ± 0.3 (3.4%)	2.47 ± 0.5 (6.9%)	2.53 ± 0.8 (8.3%)

^(a)^ Compared to vehicle control, ^(b)^ compared to diabetic control.

## Data Availability

Not applicable.

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
