# Peer review of "Phytochemical Screening, Anti-Inflammatory, and Antidiabetic Activities of Different Extracts from Caralluma edulis Plant"

_molecules, 2022, doi:10.3390/molecules27165346_

Round 1

Reviewer 1 Report

The in vitro evaluation of anti-inflammatory and in vivo antidiabetic activities of Carralluma edulis as well antioxidant capacity is very intersting. However, I have some suggestions.

It is mandatory update the epidemiology and economic burden of DM in Word. In 2021, it is stimated taht 537 million people have diabetes, around 10.5% of population. Besides, healfh expenditures due to diabetes is USD 966 billion in 2021. Pleasr, check IDF 2021.

The 4th paragraph is not clear. It should be rewritten. 

Which guidelines was used to evalute oral acute  toxicity? Why rabbits were choosen for this experiments. 

Human blood was used in membrana stabilization assay. So it is necessary submission and acceptance of the ethics committee. The authors did not show this process. 

Why the authors used only DPPH assay to evaluate the in vitro antioxidant capacity of extract's fraction? I suggest use one more in vitro antioxidant techinique such as ferric reducing antioxidant poweder (FRAP). 

The 13.45% reductin in blood glucose is not necessaryly substantial decrease. This results did not support this conclusion. 

The blood glucose leves was detected in fed or fasting animals? It is important to check the blood glucose and insulin levels, once that there no difference among control and diabetic groups at basal level in acute experiment (Table 4). 

It is important to add on the Figure 4 the scale used to analyse the liver image.  

Author Response

Comments of Reviewer-1

Comment 1. It is mandatory update the epidemiology and economic burden of DM in World. In 2021, it is estimated that 537 million people have diabetes, around 10.5% of population. Besides, health expenditures due to diabetes is USD 966 billion in 2021. Please, check IDF 2021.

Answer: Added (line no. 61-63)

Comment 2. The 4th paragraph is not clear. It should be rewritten.

Answer: It has been rewritten (line no. 71-82)

Comment 3. Which guidelines was used to evaluate oral acute  toxicity?

Answer: Oral acute toxicity test was performed according to the guidelines laid by OECD/OCDE 408. (added line no. 188-189)

Comment 4. Why rabbits were choosen for this experiments. 

Answer: Animal models have been used extensively to investigate the in vivo efficacy, mode of action and side effects of antidiabetic plants and their active principles. Due to the heterogeneity of diabetic conditions in man, no single animal model is entirely representative of a particular type of human diabetes. Thus, many different animal models have been used, each displaying a different selection of features seen in human diabetic states. Most experiments in diabetes are carried out on rodents (Ayele et al., 2021; Ibrahim et al., 2021), and rabbits (Alamgeer et al., 2016; Uzzaman & Ghaffar, 2017; Vieira et al., 2020).

We have selected the rabbits for this study because of the availability of resources (housing, feeding, equipment for analysis).

Ayele, A.G.; Kumar, P.; Engidawork, E. Antihyperglycemic and hypoglycemic activities of the aqueous leaf extract of Rubus Erlangeri Engl (Rosacea) in mice. Metabol Open, 2021, 11,  doi: 10.1016/j.metop.2021.100118.

Ibrahim, A.A.; Abdussalami, M.S.; Appah, J.; Umar, A.H.; Ibrahim, A.A.;  Dauda, K.D. Antidiabetic effect of aqueous stem bark extract of Parinari macrophylla in alloxan-induced diabetic Wistar rats. Futur J Pharm Sci 7, 164 (2021). https://doi.org/10.1186/s43094-021-00303-6

Alamgeer,; Numan, M.; Raza, S. A.; Mushtaq, M.N.; Ahmad, T.; Ahsan, H.; Asif, H.; Noor, N.; Uttra, A.M.; Arshad, L. Evaluation of anti-diabetic effects of poly-herbal product "diabetic BAL" in alloxan-induced diabetic rabbits. Acta Pol. Pharm. 2016, 73(4), 967-974.

Uzzaman, R.; Ghaffar, M. Anti-diabetic and hypolipidemic effects of extract from the seed of Gossypium herbaceum L. in Alloxan-induced diabetic rabbits. Pak J Pharm Sci. 2017, 30(1):75-86.

Vieira, G.T.; de Oliveira, T. T.; Carneiro, M.A.A.; Cangussu, S.D.; Humberto, G.A.P.; Taylor, J.G.; Humberto, J.L. Antidiabetic effect of Equisetum giganteum L. extract on alloxan-diabetic rabbit. J Ethnopharmacol. 2020, 260,  doi: 10.1016/j.jep.2020.112898.

Comment 5. Human blood was used in membrane stabilization assay. So it is necessary submission and acceptance of the ethics committee. The authors did not show this process. 

Answer: Ethical certificate has been submitted in recent email.

Comment 6. Why the authors used only DPPH assay to evaluate the in vitro antioxidant capacity of extract's fraction? I suggest use one more in vitro antioxidant technique such as ferric reducing antioxidant power (FRAP).

Answer: FRAP assay is added (Line no. Methodology: 205-216, Results: Table No. 4, Line no. 364)

Comment 7. The 13.45% reduction in blood glucose is not necessary substantial decrease. This results did not support this conclusion. 

Answer: Corrected (Line no. Results: 417-419, and 425-426, Conclusions: 662-663)

Comment 8. The blood glucose levels was detected in fed or fasting animals?

Answer: The blood glucose level was detected in fasting animals. (Corrected: line no. 274 and 281)

Comment 9. It is important to check the blood glucose and insulin levels, once that there no difference among control and diabetic groups at basal level in acute experiment (Table 4). 

Answer: Corrected in Table 6 revised (line 421)

Comment 10. It is important to add on the Figure 4 the scale used to analyse the liver image.  

Answer: Added in Figure 4 (line 495)

Reviewer 2 Report

I find the article interesting and provides information on the best solvent to extract the bioactive compounds and the anti-inflammatory and anti-diabetic properties of Caralluma edulis. The methods used could be improved to identify the bioactive compounds. The article provides a lot of information on the antioxidant, anti-diabetic and anti-inflammatory properties, although more studies are needed for the results to be conclusive.

Author Response

Comments of Reviewer-2

Comment: I find the article interesting and provides information on the best solvent to extract the bioactive compounds and the anti-inflammatory and anti-diabetic properties of Caralluma edulis. The methods used could be improved to identify the bioactive compounds. The article provides a lot of information on the antioxidant, anti-diabetic and anti-inflammatory properties, although more studies are needed for the results to be conclusive.

Answer: Thank you for your valuable comments. Some phytochemicals are identified by Spectrophotometer and data are presented in Table 3. (Line no. 346)

Authors would like to pay token of gratitude to the honorable reviewer(s) for their valuable suggestions for the improvement of the manuscript according to journal standard.

Round 2

Reviewer 1 Report

After revision, this manuscript has quality enough to be publish. 

All suggestions were accepted and the quality of the manuscript increased, including updating of epidemiology of DM.